



# Rapid quantitative analysis of SVOCs in indoor surface film using Direct Analysis in Real Time mass spectrometry: A case study on phthalates

Ying Zhou[1,2], Longkun He[1], Jiang Tan[3], Jiang Zhou[3,*], Yingjun Liu[1,4,*]

[1] SKL-ESPC & SEPKL-AERM, College of Environmental Sciences and Engineering, Peking University, Beijing, 100871, China
[2] West China School of Public Health and West China Fourth Hospital, Sichuan University, Chengdu, Sichuan 610041, China
[3] Beijing National Laboratory for Molecular Sciences, Analytical Instrumentation Center, College of Chemistry and Molecular Engineering, Peking University, Beijing, 100871, China
[4] Center for Environment and Health, Peking University, Beijing 100871, China

*Correspondence to*: Yingjun Liu (yingjun.liu@pku.edu.cn), Jiang Zhou (zhoujiang@pku.edu.cn)

**Abstract.** Direct Analysis in Real Time Mass Spectrometry (DART-MS) is recently emerged as a promising approach for measuring semi-volatile organic compounds (SVOCs) in indoor surface films. However, its broader application in indoor environments is limited by low measurement repeatability and no separation of isomers. Herein we developed a sampling suite of indoor surface film for DART-MS analysis, optimized settings of DART to obtain higher analytical performance, and demonstrated the possibility of separating isomeric compounds using tandem mass spectrometry (MS/MS). Two pair of isomeric phthalate esters, including di(2-ethylhexyl) phthalate (DEHP) and di-n-octyl phthalate (DnOP), diisobutyl phthalate (DiBP) and di-n-butyl phthalate (DnBP), were used as examples for method optimization and validation. Under optimized conditions, the instrument responses for all four compounds exhibited good linearity (r > 0.992) and acceptable repeatability (intraday RSD < 11.0%). The limits of quantification for the four PAEs ranged from 0.042 to 0.24 ng/cm$^2$. The uncertainty in the separation of isomeric components using MS/MS was <11.4%, which is acceptable for real sample analysis. To further assess the developed method, we analyzed 10 film samples collected side by side in an occupied office. DnOP was not detected. The RSD among samples was 6.1% for DEHP, 4.6% for DnBP, and 10.4% for DiBP, indicating overall good repeatability of the collection and measurement method developed. With improved performance, the developed method increases the feasibility of the DART-MS technique for monitoring dynamics of chemical composition of indoor surface films.

## 1. Introduction

People typically spend around 90% of their lives indoors (Klepeis et al., 2001; Hussein et al., 2012), making indoor exposure to air pollutants a significant concern. Indoor spaces have high a surface area to volume ratio, usually ranging from 2 to 4 m$^2$/m$^3$ (Manuja et al., 2019), which is hundreds of times greater than that outdoors. This feature enhances the importance of indoor surfaces in determining the concentrations and composition of indoor air pollutants (Ault et al., 2020). Organic films are ubiquitous on indoor surfaces (Weschler and Nazaroff, 2017). The nanometer-thick films are formed from the partitioning of semi-volatile organic compounds (SVOCs) emitted from various indoor sources, particularly those with logK$_{oa}$ values ranging from 10 to 13 (Weschler and Nazaroff, 2017; Eichler et al., 2019). Modeling analysis suggests that these surface films are important reservoirs of indoor SVOCs (Weschler and Nazaroff, 2008), and this point is supported by recent observations of indoor airborne SVOC dynamics under real-world conditions (Li et al., 2022; Lunderberg et al., 2020). However, this evidence is indirect, based solely on airborne concentration dynamics. Direct evidence showing concurrent changes in SVOC concentrations within the films themselves is still lacking.

A method for time-resolved monitoring of SVOCs in surface films is essential for exploring the surface-mediated dynamic behaviour of SVOCs. The most common approach for quantitative analysis of SVOCs in indoor surface films involves wiping existing impermeable indoor surfaces, typically glass windows, using pre-cleaned Kimwipes (Liu et al., 2003; Huo et al., 2016;



Cetin and Odabasi, 2011; Butt et al., 2004; Duigu et al., 2009) or Twillwipes (Bennett et al., 2015) wetted with organic solvents. This is followed by Soxhlet extraction or liquid-liquid extraction and gas/liquid chromatography-mass spectrometry analysis. Typically, a surface area of 0.5-2 $m^2$ is required to collect one sample, making the total area of glass windows in a specific environment the primary limiting factor of the number of samples that can be collected, and consequently, the temporal

resolution of analysis. Due to this limitation, previous studies on the growth of indoor organic films often sequentially collected 5-6 samples over a period of 1-2 months (Huo et al., 2016; Li et al., 2010; Pan et al., 2012). The largest sample size reported to date was from a study in a university hall with glass walls, where 22 samples were collected over 80 days (Huo et al., 2016). An alternative approach to measure SVOCs in surface films involves the use of highly sensitive Direct Analysis in Real Time Mass Spectrometry (DART-MS), which is pioneered by Abbatt and colleagues (Lim and Abbatt, 2020). In their approach,

SVOCs were deposited or passively collected on glass capillary tubes sealed at one end (Zhou et al., 2015; Lim and Abbatt, 2020). Subsequently, DART-MS desorbed and ionized the SVOCs on the capillary by a heated glow-discharge plasma in helium, followed by MS analysis (Cody et al., 2005; Habe and Morlock, 2015). The small size of glass capillaries allows for the deployment of tens of them in an indoor experiment, facilitating the monitoring dynamics of SVOCs in films at a higher time resolution (Lim and Abbatt, 2020). Moreover, using DART-MS to analyse indoor organic films enables rapid analysis of

samples without the need of extensive sample preparation or solvent use. However, in previous studies the measurement variability of DART-MS ranged from ±20% to ±115% without internal standards (Zhou et al., 2016; Morlock and Ueda, 2007), thus limiting its capability for quantitative analysis. This large uncertainty mainly stems from the fact that DART-MS analyses samples at atmospheric pressure in open laboratory environments, making the measurement results highly sensitive to instrumental conditions. For instance, the response of certain compounds could be increased by 50% when the temperature of

helium gas increased from 250 °C to 300°C (Morlock and Ueda, 2007). Moreover, the distance between samples, cone voltage, moving speed of the motorized linear rail, and other parameters might also impact instrument performance. However, the discussion of these influences remains incomplete. Optimizing the instrumental conditions could potentially enhance the precision of measurements.

    Moreover, due to the absence of pre-separation such as using gas/liquid-chromatography, the common combination of DART

and time-of-flight mass spectrometer is unable to distinguish isomeric compounds. This limitation further restricts the applicability of DART-MS in real-world indoor conditions, particularly considering the prevalence of various isomeric SVOCs, such as phthalate esters (PAEs), polycyclic aromatic hydrocarbons (PAHs), and polybrominated diphenyl ethers (PBDEs) (Gaspar et al., 2014; Shi et al., 2018; Li et al., 2019). For example, diisobutyl phthalate (DiBP) and di-*n*-butyl phthalate (DnBP) are among the most commonly detected SVOCs indoors (Pelletier et al., 2017). Separation of isomeric components might be

crucial in some occasions due to their varying levels of toxicity.

    The present study aims to develop a DART-MS method for fast and reliable measurements of SVOCs in indoor surface films, using two pairs of isomeric PAEs-specifically DiBP/DnBP and di(2-ethylhexyl) phthalate (DEHP) / di-n-octyl phthalate (DnOP)-as examples. A sampling suite was designed, and operating parameters of DART were optimized to improve measurement precision. In addition, DART was coupled with a triple quadrupole mass spectrometer to distinguish the isomeric

compounds. The method was further validated by analyzing the surface film samples collected from real indoor environments.

## 2. Materials and methods

### 2.1 Chemicals and Reagents

HPLC-grade acetonitrile (Fisher Scientific, Ottawa, ON, Canada) and methanol (J.T Baker, Center Valley, PA, U.S.A) were purchased and used without further purification. The standard solutions (100 mg/L in acetonitrile) of DiBP, DnBP, DEHP, and

DnOP were purchased from AccuStandard Inc. (New Haven, CT, U.S.A) with purity equal to higher than 99.8%. Palmitic acid was purchased from Dr. Ehrenstorfer GmbH (Augsburg, Germany). All solutions were prepared in acetonitrile or methanol in





an identical manner and were stored at -4 °C. Deionized water was produced by a Milli-Q purification system (Millipore, USA). The molecular structures and physiochemical properties of four phthalates studied are shown in Table 1.

## 2.2 Collection of Surface film

For film collection in real indoor environments, a sampling suite was designed, consisting of a Dip-it glass rod, a PTFE stopper plug, and a glass tube. The commercial Dip-it glass rods were employed as the substrate for film analysis. The glass rods feature a tapered design, with a long, thin tip (1.6 mm in diameter) for DART analysis and a larger diameter (5 mm) at the opposite end (cf. Figure 1). The larger end of the rod was inserted into the central hole of PTFE stopper plug (14/20 inner joint), with a small piece of PFA tubing for better sealing. During sample collection, the plug was positioned on a flat surface,

securely holding the rod with the thin tip facing upward (Figure 1-Left). SVOCs deposited on the surface of the tip, gradually forming surface film. Post-sampling, the glass rod was encapsulated in a glass tube (14/20 inner joint, Synthware, China) through the plug, ensuring that the tip, loaded with samples, did not contact any other surfaces during storage and transport (Figure 1-Right). The glass tubes were placed in the sampling environment alongside the glass rod, enabling the inner surface of the glass tube to be coated with the same surface films. This approach minimizes SVOCs repartitioning inside of the glass

tube during storage and transport, preserving the composition of the film on the sampling tip. Prior to use, all components of the sampling suites were ultrasonically cleaned with deionized water and acetonitrile. In addition, the glass rods and the glass tubes were baked for 5.5 h at 550°C to eliminate residual organics.

For application of the method, surface film samples were collected in an office in Beijing, China. A total of 10 sampling suites were placed in the office for a period of 2 months (from June to August) to collect surface films. After sampling, the sampling

suite was stored at 4°C until analysis. Samples were analyzed under quality assurance protocols, including double method blanks, a field blank, double solvent blanks, and two samples for calibration.

## 2.3 Preparation of spiked samples for testing and calibration

To prepare spiked samples, we first applied 5 μL of a palmitic acid solution, a prominent component of indoor surface film (Weschler and Nazaroff, 2017), to the tip of a clean glass rod using a micro-pipette. The solution was then allowed to dry,

forming a base film of palmitic acid. Next, 5 μL solutions of target PAEs in acetonitrile were applied and allowed to dry. Pre-treating the glass rods with less volatile palmitic acid provides a matrix for the more volatile phthalates to dissolve into, thereby preventing their evaporation after application. The spiked samples were analyzed immediately following the preparation.

## 2.4 DART-MS/MS analysis

Analyses were conducted using DART-SVP ionization source (IonSense, Saugus, MA, USA) coupled with a triple quadrupole

tandem mass spectrometer (MS/MS; 5500, SCIEX, Framingham, MA, USA). The ion source was operated in positive ion mode with high-purity helium (He, 99.999%) as the reagent gas (3.0 SLM). The potential applied to the gas-discharged needle was 6 kV. The grid electrode voltage was 350 V. Given that the boiling points of the studied PAEs ranged from 220 to 384 °C (Table 1), the He flux temperature of 150°C, 200 °C, 250 °C, and 300 °C was tested. In the interface of mass spectrometer, a ceramic tube (3.18 mm i.d., 79 mm in length) was used as an ion inlet. The distance between the DART source and the ceramic

tube was minimized to 10 mm to favor the desorption of the target SVOCs and to enhance ions transfer to MS/MS.

During analysis, the Dip-it glass rods were placed on the Dip-it sample holder (IonSense, Saugue, MA USA) mounted on the motorized linear rail (IonSense, Saugue, MA USA), as shown in Figure S1. The holder can accommodate up to 12 glass rods, with 9 mm in between. The moving speed of the rail was controlled by a computer program (DART-SVP control software version 3.2.2). When the rod passed between the ceramic tube and the DART ion source, the heated plasma gas flow directly

desorbed organics on glass rods toward the sampling orifice of the mass spectrometer (Cody et al., 2005). This process can lead to a discernible peak in the time series of corresponding ion signals recorded by DART. Herein the resulting peak area





was used for quantitative analysis of sorbed organics. The desorption area for DART source was measured to be 7.4 mm in diameter, by heating a piece of polymeric plate at the highest He flux temperature (Hayeck et al., 2015). When placed on the holder, the tip of the glass rod was 0.4 mm below the centre of the DART source, resulting in an effective desorption area of

0.24 cm$^2$ on the glass rod.Moving speed of the rail was tested under 0.2, 0.5, 0.7 mm/s.

Parameters of the triple quadrupole mass spectrometer were optimized using standard solutions of targeted species using conventional electrospray ion (ESI) source. The spectrum was recorded in a positive mode. PAEs exhibited distinct protonated ions [M + H]$^+$, which were used as precursor ions for analysis. The major products ions for individual compounds were identified through triple-quadrupole scanning. The optimized condition for individual precursor/product ion pairs, including

declustering potential (DP) and collision energy (CE), is shown in Table S1. Mass calibration of the MS/MS was performed daily to ensure the accuracy of mass analysis.

### 3. Results and discussion

#### 3.1 Product ion pattern of PAEs in triple quadrupole mass spectrometer

Figure 2 compares the pattern of major product ions for the two pairs of isomeric PAEs. The ion intensities were normalized

to that of the strongest product ion for individual compounds. For both DEHP and DnOP (precursor ions at *m/z* 391.3), the strongest product ion appeared at *m/z* 149.1. Products ions also appeared at *m/z* 279.1, 261.1, 167.2, and 113.0, but their relative signals differed for the two compounds. For example, the fragment ion at *m/z* 167.2 was a few times more abundant for DEHP than that for DnOP, while the fragment ion at *m/z* 261.1 was more abundant for DnOP than that for DEHP. Similarly, for DnBP and DiBP (precursor ions at *m/z* 279.2), the strongest product ion both occurred at *m/z* 149.1, but the relative signals of

other product ions differed. In particular, the product ion at *m/z* 57.3 was almost exclusively found from DiBP rather than DnBP. In the following analysis, we used the strongest product ions for method optimization (section 3.2). For method validation and real sample analysis, the signals of multiple product ions were measured to separate isomeric PAEs (sections 3.3 and 3.4).

#### 3.2 Optimization of DART settings

Figure 3 shows the relative peak intensities of four PAEs at He flux temperature of 150 °C, 200 °C, 250 °C, and 300 °C. The intensities were quantified by integrating the peak area on the time series of respective quantitative ion pairs. As shown in Figure 3, the strongest response occurred at 200°C for all four PAEs studied. At 150°C, the signals of less volatile DEHP and DnOP were only <4.5 % of those at 200 °C. At 300°C, the signals of more volatile DiBP and DnBP dropped to <8.5 % of those at 200 °C. These results suggest that the transmission efficiency of SVOCs is highly sensitive to the He flux temperature.

A lower temperature (e.g., 200 °C) might have led to inadequate desorption of PAEs from the glass rods, while at higher temperature (e.g., 300 °C) volatilization and desorption of PAEs was too rapid so that a reduced fraction of vaporized PAEs was ionized and transferred to the mass spectrometer. Additionally, as the temperature increased from 200°C to 300°C, the peaks on the time series exhibited broadening, accompanied by the emergence of small negative peaks at the tail (Figure S2). In the following analysis, 200 °C was adopted.

The instrument performance was also influenced by the distance between adjacent glass rods and the moving speed of the metal holder. As shown in Figure 4, using the default 9-mm distance resulted in an elevated baseline on the time series, indicating overlaps of PAE signals on adjacent glass rods. With the distance of 18 mm between adjacent glass rods, the peak shapes became symmetric and the baseline was reduced to the background level, allowing for separation of PAE signals on adjacent rods. Given the limited length of the motorized linear rail, further increase of the distance can reduce the number of

glass rods analyzed in one batch, and thus an 18-mm distance was used herein. Moreover, three different moving speeds, 0.2, 0.5, and 0.7 mm/s, were tested (Figure 5). The best performance occurred at the speed of 0.5 mm/s. The recoveries decreased



at higher or lower speeds. Lower speed resulted in broader peaks and caused unstable peak shape, while higher speed induced insufficient ionization on each sample. Thus, the speed of 0.5 mm/s was adopted in the current analysis.

**3.3 Method evaluation using spiked samples**

(1) Calibration curves

As shown in Table 2, calibration curves were obtained at five surface concentrations ranging from 0 to 1.20 ng/cm² for DEHP and DnOP, 0 to 0.85 ng/cm² for DiBP and DnBP. Calibration curves were linear over the spiked range with the four PAEs with correlation coefficients over 0.98 (Table S2). Higher surface concentration could lead to saturation of signals of some ion pairs such as $m/z$ 391.3/149.1 for DEHP and DnOP. The limits of detection (LODs) and and limits of quantification (LOQs)

for individual PAEs were respectively calculated based on three times and ten times the signal-to-noise (peak to peak) ratio of their major product ions at the lowest standard concentrations. The obtained results were listed in Table 2. The LOQ of DEHP was 0.042 ng/cm². Compared to the reported DEHP concentration in indoor surface film, e.g., 0.23 ng/cm² (summer) or 0.45 ng/cm² (winter) after initial 7 days of growth (Huo et al., 2016), the LOQ was one order of magnitude lower. The implication is that this method might allow for tracking the growth of surface film in the time resolution of one to several days. The LOQ

of DiBP and DnBP were 0.24 and 0.085 ng/cm², respectively. The substantially higher LOQ of DiBP compared to other tested phthalates is mainly due to elevated indoor air concentration of DiBP in the laboratory.

(2) Repeatability

To determine the instrument and the methodology repeatability, measurements at two spiked levels were repeated three times a day (n=15) and on three consecutive days for individual tested phthalates (n=15). Precision was calculated as the relative

standard deviation (RSD, %) for intraday and interday repetitions. As shown in Table 3, for MS/MS alone, the intraday RSD ranged from 3.7% to 8.1% for four PAEs while the interday RSD ranged from 3.9% to 7.0%. The method repeatability (using DART-MS/MS as a whole) was inferior to the MS/MS repeatability for all compounds, but to a reasonable extent. The intraday RSD ranged from 6.7% to 11.0%, and the interday RSD ranged from 7.3% to 18.8%. The RSDs of MS/MS repeatability tests were around 74% (±21%) of those in method repeatability tests, which reveals that the method instability herein mainly rose

from the fluctuations of MS/MS instead of DART source, in particular for DnBP and DiBP. The obtained method RSD in this study is significantly lower than that in previous studies, which indicated 20% (Zhou et al., 2016) and 115% (Morlock and Ueda, 2007) for intraday.

(3) Separation of isomeric compounds using DART-MS/MS

Herein the concentrations of isomeric PAEs were estimated based on different product ion patterns of isomeric PAEs using

the following linear system of equations:

$$R = S \times C$$

where R is a column vector of acquired responses of product ions, S is a matrix of the product ion sensitivities of isomeric PAEs which were obtained through calibration presented earlier, and C is a column vector of concentrations of isomeric PAEs. In the case that the number of product ions is greater than the number of isomeric compounds, the above linear system is

overdetermined and a least square solution for concentration vector C is obtained. Herein only product ions with intensities greater than 8% of the strongest product ion for at least one isomer were used for quantification. Among the product ions for each set of isomeric PAEs shown in Figure 2, all five ions were used to separate DEHP and DnOP, while three ions ($m/z$ 205.0, $m/z$ 149.1, and $m/z$ 57.3) were used to separate DiBP and DnBP.

To verify the accuracy of this separation method, spiked samples with different concentration ratios of isomeric pairs (1:1, 2:1,

and 1:2) for DEHP/DnOP and DnBP/DiBP were prepared and analyzed using DART-MS/MS. Table 4 shows the obtained concentrations of the spiked mixed standards of PAE isomers. The results were close to the spiked concentrations with relative





errors ranged from 0.8% to 11.4% for the four PAEs. This result suggests that the developed method effectively separates the
mixture of isomers.

### 3.4 Application to real indoor surface film samples

The DART-MS/MS method developed was further assessed by analyzing film samples collected in real indoor environment. Ten clean glass rods were exposed simultaneously in an office for 60 days to allow growth of films on their surfaces. Table 5 summarizes the measurement results of the four concerned PAEs. DEHP, DiBP and DnBP were detected in all samples, whereas DnOP was not detected. This result is consistent with some previous measurements of PAEs in indoor air (Takeuchi et al., 2014; Takeuchi et al., 2018; Huang et al., 2021). A possible reason might be relatively low usage of DnOP in products

in the office. After 60 days of growth, the mean concentration of DEHP in the surface film was 1.045 ng/cm$^2$, which is similar to the concentrations of 1.3 ng/cm$^2$ detected in window film after 77 days of growth days in Harbin, China (Huo et al., 2016). The mean concentration of DnBP and DiBP were similar, which were 0.739 ng/cm$^2$ and 0.631 ng/cm$^2$, respectively. The RSD among samples was 6.1% for DEHP, 4.6% for DnBP, and 10.4% for DiBP, indicating overall good repeatability of the collection and measurement method employed.

### 4. Summary

This study presents a sensitive DART-MS/MS method for the fast and accurate quantification of SVOCs in organic films without the need for pre-treatment. Compared to previously reported DART-MS method for this purpose, this method developed herein offers substantially improved repeatability in the absence of internal standards. In addition, by utilizing MS/MS analysis, separation of isomeric components within films becomes possible. These developments increase the

feasibility of the DART-MS approach for studying the dynamics of SVOCs in indoor surface film. However, there are still some limitations associated with this method. Due to the small sampling area (~0.24 mm$^2$), the film mass is hard to be measured gravimetrically, unlike traditional methods using ~1-m$^2$ surface-wipe samples. Consequently, surface concentrations rather than volume concentrations are obtained. Furthermore, MS/MS analyisis is suitable only for targeted analysis for a limited number of compounds, and maintaining time resolution might be challenging for an extensive list of targeted compounds.


*Data availability.* All data can be provided by the authors upon request.

*Author contributions.* The authors ZY and HLK were responsible for the sample collection of the study. Data collection, data analysis, and the methodology development were led by ZY with the help of TJ, under the supervision of LYJ and ZJ. The

method for separating and evaluating the data of isomeric components was written by ZY and HLK. The original draft of the paper was written by ZY. The reviews of the paper were done by LYJ and ZJ.

*Competing interests.* The contact author has declared that none of the authors has any competing interests.

*Financial support.* This research has been supported by the Natural Science Foundation of China (grant no. 22376003) and the National Key Research and Development Program of China (grant no. 2023YFC3710100).



**Table 1.** Physiochemical properties of the studied phthalates

| Compounds | Abbr. | CAS. No | Molecular formula | $logK_{oa}$ | Boiling point (°C) | Vapor pressure (Pa, 25°C) |
|---|---|---|---|---|---|---|
| di(2-ethylhexyl) phthalate | DEHP | 117-81-7 | $C_{24}H_{38}O_4$ | 12.9 [a] | 384 [b] | $1.30\times10^{-5}$ [c] |
| di-$n$-octyl phthalate | DnOP | 117-84-0 | $C_{24}H_{38}O_4$ | 12.1 [b] | 435 [d] | $1.30\times10^{-5}$ [c] |
| diisobutyl phthalate | DiBP | 84-69-5 | $C_{16}H_{22}O_4$ | 9.62 [a] | 296.5 [b] | $4.73\times10^{-3}$ [e] |
| di-$n$-butyl phthalate | DnBP | 84-74-2 | $C_{16}H_{22}O_4$ | 9.83 [a] | 340 [b] | $3.60\times10^{-3}$ [c] |

[a] Weschler and Nazaroff (2010);
[b] US Environmental Protection Agency's (USEPA) EPI Suite™;
[c] Wang et al. (2008);
[d] Wu et al. (2016);
[e] Feng et al. (2020).

**Table 2** Linear range and LOD/LOQ for the four selected PAEs spiked on glass rods.

| Compounds | Quantitative ion pairs ($m/z$) | Linear range (ng/cm²) | LOD (ng/cm²) | LOQ (ng/cm²) |
|---|---|---|---|---|
| DEHP | 391.3/149.1 | 0.04~1.20 | 0.014 | 0.042 |
| DnOP | 391.3/149.1 | 0.09~1.20 | 0.028 | 0.089 |
| DiBP | 279.2/205.0 | 0.24~0.85[a] | 0.072 | 0.24 [a] |
| DnBP | 279.2/205.0 | 0.085~0.85[a] | 0.025 | 0.085 |

[a] The lower limit of linear ranges were elevated for DiBP due to its high background levels in the indoor air of the analytical lab.

**Table 3** Interday and intraday repeatability for the four investigated PAEs.

| Compounds | Precursor ions ($m/z$) | Product ions [a] ($m/z$) | Spiked level (ng/cm²) | MS/MS repeatability (RSD%) (n=15) | | Method repeatability (RSD%) (n=15) | |
|---|---|---|---|---|---|---|---|
| | | | | Intraday | Interday | Intraday | Interday |
| DEHP | 391.3 | 149.1/167.2/113.0 | 0.085 | 5.0~7.1 | 4.7~7.0 | 6.7~9.3 | 9.9~18.8 |
| | | /279.1/261.1 | 0.637 | 3.7~5.7 | 3.9~6.0 | 8.8~9.8 | 10.1~17.7 |
| DnOP | 391.3 | 149.1/261.1 | 0.085 | 6.6~8.0 | 7.0~6.1 | 6.7~10.7 | 8.3~13.1 |
| | | (113.0/167.2/279.1) | 0.637 | 6.9~7.7 | 6.3~6.7 | 9.5~11.0 | 11.2~13.8 |
| DiBP | 279.2 | 205.0/149.1/57.3 | 0.24 | 5.4~6.9 | 5.4~7.0 | 8.4~9.1 | 7.8~8.6 |
| | | (223.1/167.1) | 0.637 | 4.5~4.7 | 6.0~6.5 | 8.0~10.5 | 7.3~9.3 |
| DnBP | 279.2 | 205.0/149.1 | 0.085 | 6.9~8.1 | 5.8~6.6 | 6.8 | 8.2 |
| | | (223.1/167.1/57.3) | 0.637 | 6.8~8.0 | 4.9~6.7 | 10.5 | 10.8 |

[a] The signal intensities of these product ions in the parentheses were below 8% of the highest detected signal intensity, thus these ions were excluded in the calculation of instrument repeatability and method repeatability.





**Table 4** The validation of the method for the separation of PAE isomers by using DART-MS/MS.

| Mixed standard | Compounds | Spiked conc. (ng/cm$^2$) | Calculated conc. (ng/cm$^2$) | Mixed standard | Compounds | Spiked conc. (ng/cm$^2$) | Calculated conc. (ng/cm$^2$) |
|---|---|---|---|---|---|---|---|
| DEHP+DnOP (1:1) | DEHP | 0.297 | 0.310 | DnBP+DiBP (1:1) | DnBP | 0.297 | 0.307 |
|  | DnOP | 0.297 | 0.267 |  | DiBP | 0.297 | 0.277 |
| DEHP+DnOP (1:2) | DEHP | 0.297 | 0.299 | DnBP+DiBP (1:2) | DnBP | 0.297 | 0.301 |
|  | DnOP | 0.594 | 0.668 |  | DiBP | 0.594 | 0.591 |
| DEHP+DnOP (2:1) | DEHP | 0.594 | 0.635 | DnBP+DiBP (2:1) | DnBP | 0.594 | 0.614 |
|  | DnOP | 0.297 | 0.283 |  | DiBP | 0.297 | 0.286 |

**Table 5** The concentrations (ng/cm$^2$) and detection frequency of film samples collected in an office (n=10).

| Compound | DF[a] (%) | Mean | GM[b] | Range | RSD (%) |
|---|---|---|---|---|---|
| DEHP | 100 | 1.045 | 1.044 | 0.947 ~ 1.195 | 6.1 |
| DnOP | 0 | n.d. [c] | n.d. [c] | n.d.[c] | - |
| DnBP | 100 | 0.739 | 0.745 | 0.685~0.788 | 4.6 |
| DiBP | 100 | 0.631 | 0.634 | 0.502~0.735 | 10.4 |

[a] DF: detection frequency; [b] GM: geometric mean; [c] n.d.: below the limit of detection.



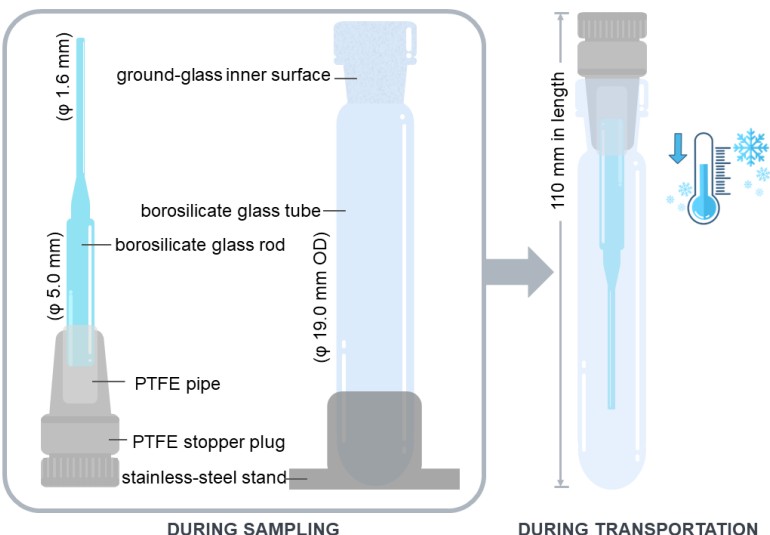

**Figure 1:** Schematic of indoor surface film passive sampling suite.

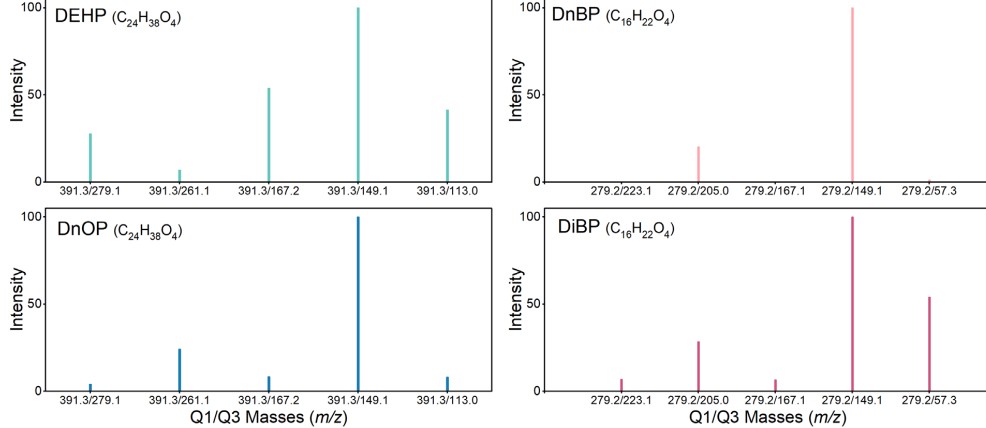

**Figure 2:** Relative intensity of product ions of DEHP/DnOP at *m/z* 391.3 (left) and DnBP/DiBP at *m/z* 279.2 (right).



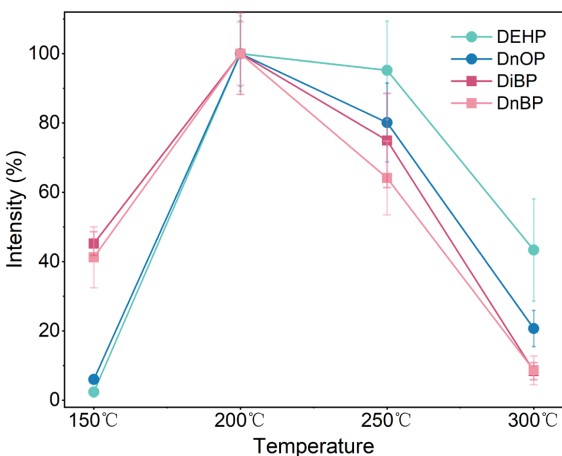

**Figure 3:** Relative intensities for 0.05 ng spiked DEHP, DnOP, DiBP and DnBP at different He flux temperature.

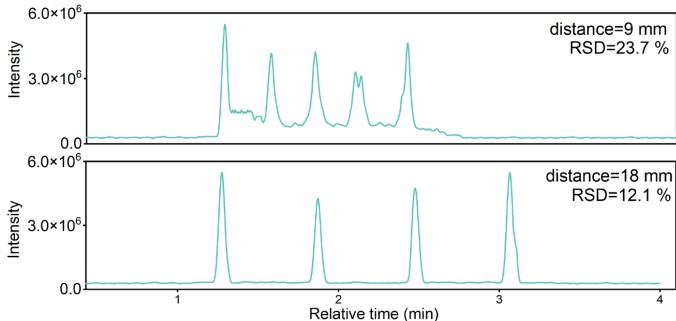


**Figure 4:** The influence of rod distance on the total ion chromatograms (TIC) for spiked DEHP. The He flux temperature was set as 200°C. The speed of motorized linear rail was 0.5 mm/s.

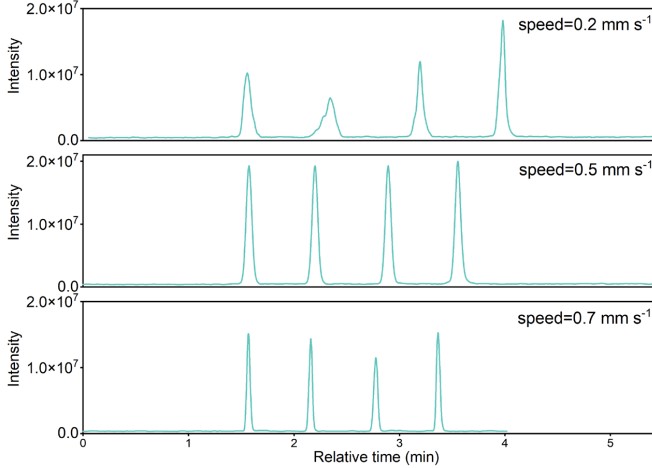

**Figure 5:** The influence of speeds of motorized linear rail on the total ion chromatograms (TIC) for spiked DEHP within 5

minutes. The He flux temperature was set as 200°C. The rod distance was 18 mm.



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
