# Peer review of "Rapid quantitative analysis of SVOCs in indoor surface film using Direct Analysis in Real Time mass spectrometry: A case study on phthalates"

_Atmospheric Measurement Techniques, 2024_

## Author Response (AR1)

We appreciate the reviewer for providing us useful comments. In the following, original reviewer comments, author's responses, and corresponding updates on the main text.

**Comments from Reviewer #1**

In this study, a rapid quantitative analysis of PAEs in indoor surface film using direct analysis in real time mass spectrometry was developed. The experiment was well organized, and the method was comprehensively validated, which provided a new method for the analysis of SVOCs in indoor environment.

Thank you for the positive comments on our manuscript. Responses to the individual comments are given below.

1. **What's the meaning for DART? Without pretreatment? How about the real time to be achieved?**

Thank you for the questions. DART is an ionization technique termed Direct Analysis in Real Time (DART) in mass spectrometry. DART-MS can be applied for rapid analysis of a wide variety of samples without pretreatment, such as tablets, bodily fluids, polymers, glass, plant leaves, fruits & vegetables, at atmospheric pressure and in the open laboratory environment. The feature of requiring no sample pretreatment allows for real-time analysis for some purposes, such as analysis of explosives or drugs. In this study, we did not make real-time (or online) measurement of surface film. Instead, we collected the surface film and made off-line measurements using DART-MS.

2. **For the sampling collection, how many days should be used for the sampling?**

Thank you for the question. For phthalates, according to LOQs in our developed method and reported growth of phthalate concentrations in surface films, sampling times down to several day should be feasible. That said, the sampling time should be determined based on the specific SVOCs concentrations in the air. We have discussed this in Line 175-178 in the original manuscript. The text has been revised as below to improve the clarity:

*"The LOQ of DEHP was 0.042 ng/cm². Compared to the reported DEHP concentration in indoor surface film, e.g., ~0.13 ng/cm² (summer) or ~0.37 ng/cm² (winter) after initial 7 days of growth (Huo et al., 2016), the LOQ was several times lower. The implication is that this method might allow for tracking changes of DEHP concentration during the growth of surface film in the time resolution of one to several days."*

3. **For the influence of He flux temperature, the author indicated that the temperature of 200 can lead to the inadequate desorption, why the temperature was used finally?**

Thank you for your correction. In line 151, we intended to indicate that the lower temperature of 150 °C, but it was incorrectly written as 200 °C in the manuscript. We have now corrected this in the revised version.

4. **For the calculation of LOD and LOQs, the spiked samples were used. How can the indoor air concentration of DiBP influence the value of its LOQ (Line 175-176)?**

Thank you for your input. The LOD and LOQ were defined as the minimal amounts of analyte detected with a signal-to-noise ratio of 3 and 10 times, respectively. As illustrated in raw time series plots for spiked DEHP sample

(Figure R1), the noise level was determined from baseline measurements taken between sample measurements of glass robs. Since DART is an atmospheric pressure ionization technique with an open source, the baselines were influenced by the concentration levels of compounds present in the laboratory air. In this study, the concentrations of DiBP in the laboratory was much higher than that of other PAEs, which resulted in an elevated baseline noise level of DiBP.

To clarify the idea, we have revised the text as below:

*"The limits of detection (LODs) and limits of quantification (LOQs) for individual PAEs were calculated based on three and ten times the signal-to-noise (peak to peak) ratio, respectively, using their quantitive ion pairs at the lowest standard concentrations. The noise levels were determined from baseline measurement taken between sample measurements during the movement of the sample holder (i.e., measurements between signal spikes, as shown in Figure S2), and these noise levels are largely influenced by the composition of the laboratory air."*

[Figure]

Figure S1: Illustration of raw time series data for spiked DEHP samples.

**5. Line 190, for the equation, the reference should be added.**

Thanks for your advice. This formula was developed specifically for this study to separate the isomeric compounds with DART-MS/MS, so there are no references.

**6. Line 210-211, the direct comparison between the study and the previous study should be carefully, because the two types of the indoor rooms were different.**

Thanks for your suggestion. We have revised the sentence to clarify the differences in indoor environments where the two measurements were conducted. Our intention here is simply to compare the DEHP concentrations we measured with those reported in other studies, as a rough assessment whether our results are reasonable. This comparison is not intended to lead to a generalized conclusion. The revised text now reads:

*"After 60 days of growth, the mean concentration of DEHP in the surface film was 1.045 ng/cm² , which is comparable to the concentrations of ~0.46 ng/cm² (summer) or~1.89 ng/cm² (winter) measured in window film*

*after 60 of growth days in a university hall in Harbin, China (Huo et al., 2016)."*

**7. Did the author use the method for testing other SVOCs in indoor environment?**

Thank you for your question. This study aimed to enhance the measurement repeatability of DART-MS analysis for surface film samples and to investigate the potential for separating isomeric compounds, with phthalates serving as a case study. To achieve this, MS-MS was employed in selective ion mode (SIM), focusing exclusively on the mass spectral signals of targeted phthalate ions. In subsequent research, we utilized a different high-sensitivity mass spectrometry technique, FT-MS, to monitor a broader array of SVOCs under the DART conditions optimized in this study. The scientific results of these investigations will be detailed in our future publications. The methods presented here lay a technical foundation for our ongoing scientific exploration.

There are some writing problems for correction:
**8. In abstract, two pair should be two pairs.**

Thank you for your suggestion. It has been revised.

**9. In line 174, the LOQ should be the LOQs.**

Thank you for your suggestion. It has been revised.

We appreciate the reviewer for providing us useful comments. In the following, original reviewer comments, author's responses, and corresponding updates on the main text.

**Comments from Reviewer #2**

Zhou et al. optimized the method to quantify phthalate esters (PAEs) on glass rods with Direct Analysis in Real Time mass spectrometry (DART-MS), specifically, for two pairs of isomeric PAEs, including DEHP and DnOP, DiBP and DnBP. They developed a method to distinguish isomers based on MS/MS information. They also collected 10 film samples and subsequently analyzed with the DART.

We thank the reviewer for the comments. Responses to the individual inputs are given below.

1. The paper is written clearly; however, it is quite short with only lab optimization and 10 film samples. My major comment is about the overall applicability of the method to other types of SVOCs, given that the authors' aim is to develop a method to quantify SVOCs on surfaces as written in the title and introduction. They tested only 4 chemicals, and the applicability of the method to other species are not guaranteed, especially for those with different physical chemical properties, and less abundant species indoors (PAEs are typically quite high in most indoor environments). For example, for the paragraph on line 145, the optimization is specific to PAEs (e.g., temperature settings, etc., which depends on the properties of the measured species). How can these parameterization/optimization be applied for analysis of other compounds? It would be better if the authors measured a few groups of SVOCs. The 10 environmental samples actually contain other types of SVOCs, can the authors do some information mining to prove the feasibility of the method?

We thank the reviewer for this comment. This study aimed to enhance the measurement repeatability of DART-MS analysis for surface film samples and to investigate the potential for separating isomeric compounds, with phthalates serving as a case study. To achieve this, MS-MS was employed in selective ion mode (SIM), focusing exclusively on the mass spectral signals of targeted phthalate ions.

While our study specifically optimized DART parameters and operating conditions for phthalates (ranging from DiBP/DnBO to DEHP/DnOP), we believe these optimizations can be effectively applied to other SVOCs with similar vapor pressure ranges. In subsequent research, we utilized a different high-sensitivity mass spectrometry technique, FT-MS, to monitor a broader array of SVOCs under the DART conditions optimized in this study. The scientific results of these investigations will be detailed in our future publications. The methods presented here lay a technical foundation for our ongoing scientific exploration.

We have added one sentence in the Summary session to clarify the above points.

*"Although the optimizations were specifically tailored for phthalates, they may be effectively applied to other SVOCs with similar vapor pressures."*

**Specific comments:**
2. **Line 32: specify Koa**

Thank you for your suggestion. It has been revised. The revised text reads:

*"The nanometer-thick films are formed from the partitioning of semi-volatile organic compounds (SVOCs) emitted from various indoor sources, particularly those with octanol-air partition coefficient ($K_{oa}$) values ranging from $10^{10}$ to $10^{13}$ (Weschler and Nazaroff, 2017; Eichler et al., 2019)."*

3.  **Line 103: Is this pure palmitic acid or solution? If it is solution, what solvent was used and what is the concentration of palmitic acid?**

    Thanks for the advice. The 20 μg/mL of palmitic acid in methanol has been used. The concentration has been added in the manuscript.

4.  **Line 126: why is ESI used for optimizing the mass spectrometer? why not using the DART source?**

    Thanks for this comment. The ions produced from the investigated phthalates using both DART and ESI ionization methods are similar, with both generating [M+H]$^+$ ions. We optimized two key mass spectrometer parameters—declustering potential (DP) and collision energy (CE)—to fine-tune the distribution of fragment ions from these [M+H]$^+$ ions. DART, being an atmospheric pressure ionization technique with an open source, is susceptible to various external factors. In contrast, ESI offers more stable ion generation, enabling more consistent optimization of mass spectrometer performance. We have added a sentence in the revised text to clarify this point:
    *"Compared to DART, ESI produces similar product ions but provides more stable ion generation, and hence it is used for optimizing mass spectrometer performance here."*

5.  **Line 135, Figure 2: Were the signals of parent ion normalized to the signals of individual product ions? The x-axis is not a typical way to present mass spectra. Why not present in the typical way, i.e., just using m/z? The current version is not very straightforward.**

    Thanks for your advice. Figure 2 has been revised as you suggested:

[Figure]

Figure 2: Relative intensity of selected product ions of DEHP/DnOP at *m/z* 391.3 (left) and DnBP/DiBP at *m/z* 279.2 (right).

6.  **Line 175: Do the authors mean contamination of glass rod, or the instrument system?**

    Thank you for this question. It is neither a contamination of glass rod or instrument. Since DART is an atmospheric pressure ionization technique with an open source, the concentrations of compounds in the laboratory

air could influence the baselines of the ion chromatograms (the noise level). Thus, the LOD and LOQ which were defined as the three times and ten times the signal-to-noise ratio will be affected by composition of laboratory air. Please see elaboration in the response to comment (4) of reviewer #1.

7. **How high is DEHP, DiBP in lab air? If lab air contamination is problem, how can the method be applied for real time measurement as the authors propose to study?**

   Thank you for this comment. Please see detailed explanation in the response to comment (4) of reviewer #1. The main interference from lab air contamination is its effects on the intensity of the chromatogram baseline (the noise level). The method is proposed for analyzing film samples collected in indoor environments, and the baselines and signals were measured alternatively in each sample run, so it can be easily subtracted.

8. **Line 190: for genuine film samples, the composition is very complexed, and many species may have quite low abundance. Would this method work? As suggested earlier (major comment), can the authors use this method to quantify other species in the sample, other than the 4 PAEs? For example, for other species in the 10 samples.**

   Thank you for your suggestion. Please see our response to major comment (1). Since the film samples have been desorbed into the MS during analysis and those 10 samples have no duplicates, we are unable to reanalyze these films for other SVOCs using the scanning mode.